# Evaluation of Molecular Test for the Discrimination of “Naked” DNA from Infectious Parvovirus B19 Particles in Serum and Bone Marrow Samples

**DOI:** 10.3390/v14040843

**Published:** 2022-04-18

**Authors:** Arthur Daniel Rocha Alves, Barbara Barbosa Langella, Mariana Magaldi de Souza Lima, Wagner Luís da Costa Nunes Pimentel Coelho, Rita de Cássia Nasser Cubel Garcia, Claudete Aparecida Araújo Cardoso, Renato Sergio Marchevsky, Marcelo Alves Pinto, Luciane Almeida Amado

**Affiliations:** 1Laboratório de Desenvolvimento Tecnológico em Virologia, Instituto Oswaldo Cruz, FIOCRUZ, Avenida Brasil, 4365, Manguinhos, Rio de Janeiro 21040-900, Brazil; arthuralves@aluno.fiocruz.br (A.D.R.A.); babi.langella@gmail.com (B.B.L.); mari.magaldi13@gmail.com (M.M.d.S.L.); wagnercoelho@aluno.fiocruz.br (W.L.d.C.N.P.C.); marcelop@ioc.fiocruz.br (M.A.P.); 2Departamento de Microbiologia e Parasitologia, Instituto Biomédico, Universidade Federal Fluminense, Niterói, Rio de Janeiro 24210-230, Brazil; ritacubel@id.uff.br; 3Departamento Materno Infantil, Faculdade de Medicina, Universidade Federal Fluminense, Niterói, Rio de Janeiro 24033-900, Brazil; claudetecardoso@id.uff.br; 4Vice Diretoria de Qualidade, Instituto de Tecnologia em Imunobiológicos, FIOCRUZ, Avenida Brasil, 4365, Manguinhos, Rio de Janeiro 21040-900, Brazil; march@bio.fiocruz.br

**Keywords:** Benzonase^®^, parvovirus B19, acute infection, persistent infection, viral DNA load

## Abstract

Low levels of parvovirus B19 (B19V) DNA can be detected in the circulation and in different tissue of immunocompetent individuals for months or years, which has been linked to inflammatory diseases such as cardiomyopathy, rheumatoid arthritis, hepatitis, and vasculitis. However, the detection of B19V DNA does not necessarily imply that infectious virions are present. This study aimed to evaluate the method based on the Benzonase^®^ treatment for differentiation between the infectious virions from “naked” DNA in serum and bone marrow (BM) samples to be useful for the B19V routine diagnosis. In addition, we estimated the period of viremia and DNAemia in the sera and bone marrow of nonhuman primates experimentally infected with B19V. Serum samples from ten patients and from four cynomolgus monkeys experimentally infected with B19V followed up for 60 days were used. Most of the human serum samples became negative after pretreatment; however, only decreased viral DNA loads were observed in four patients, indicating that these samples still contained the infectious virus. Reduced B19V DNA levels were observed in animals since 7th dpi. At approximately 45th dpi, B19V DNA levels were below 10^5^ IU/mL after Benzonase^®^ pretreatment, which was not a consequence of active B19V replication. The test based on Benzonase^®^ pretreatment enabled the discrimination of “naked DNA” from B19V DNA encapsidated in virions. Therefore, this test can be used to clarify the role of B19V as an etiological agent associated with atypical clinical manifestations.

## 1. Introduction

Parvovirus B19 (B19V) is a small (23–28 nm in diameter), nonenveloped icosahedral virus containing a single-stranded DNA genome of 5596 nucleotides [1]. B19V is classified as belonging to the *Parvoviridae* family, *Erythroparvovirus* genus, and *Primate erythroparvovirus 1* species, due to its tropism for erythroid progenitor cells, predominantly in the bone marrow and fetal liver [1,2].

B19V is a common, global pathogen that infects children and adults, and serological studies indicate that over 40–60% of healthy adults have anti-B19V IgG antibodies due to previous exposure [3,4]. B19V is mainly transmitted via the respiratory route. Still, it can also be transmitted vertically from the mother to the fetus via the transfusion of whole blood or pooled blood products or via organ transplantation [3,4].

Viral transmission occurs most often during the viremia period that precedes the clinical presentation [3,4,5]. Approximately one week after a respiratory infection, a high amount of viral DNA (>10^12^ IU/mL) can persist for 6 to 7 days in the peripheral blood, saliva and respiratory secretions of acutely infected individuals. The classical slapped-cheek rash associated with erythema infectiosum (also known as the fifth disease) and arthralgia develops 17–18 days after infection, at the time of the appearance of IgM- and IgG-specific antibodies [4,5].

B19V infection is usually acute and self-limited and causes no symptoms in most healthy individuals [6]. However, the virus can also induces transient aplastic crisis in patients with underlying hemolytic anemia [7], persistent anemia in immunocompromised patients and hydrops fetalis and fetal loss in pregnant women [8,9]. The wide range of diseases depends on the hematological and immune status of the host [10].

B19V DNA can be detected at low levels (10^4^ IU/mL) in the blood and various tissues (bone marrow, skin, tonsils, liver and heart) of immunocompetent individuals for months or years after acute infection, even when B19V antibodies are present [11,12,13]. However, whether the detection of viral DNA in the blood and tissues of seropositive and asymptomatic individuals reflects the presence of an infectious virus has not been elucidated. This point is of particular concern since B19V is resistant to most inactivation procedures used in manufacturing blood-derived products and can be transmitted by pooled blood products [11,12,13]. Furthermore, B19V DNA persistence in these tissues may suggest a causal relationship with some clinical conditions, such as myocarditis, chronic arthropathy and acute liver failure (ALF) [3,13,14,15].

Recently, Molenaar-de Backer and collaborators [16] presented a method for differentiating between B19V DNA in EDTA-plasma samples resulting from active viral replication and B19V viremia. The method is based on plasma pretreatment with Benzonase^®^ prior to nucleic acid extraction. The authors proposed that the remaining B19V DNA can be released from tissues without active replication, suggesting that the role of B19V in clinical syndromes, such as myocarditis and arthritis, based mainly on the detection of B19V DNA in the blood and tissues without further support by serology, clinical signs or epidemiology should be carefully reconsidered [16].

A case of recurrent B19V DNAemia in a patient with hereditary spherocytosis, which was initially interpreted as viral reactivation, was clarified after confirming the absence of infectious viral particles in the patient’s blood. The hypothesis was raised that the second episode of DNAemia represented the mass release of B19V DNA from the bone marrow and was, therefore, not the result of active infection [17].

In a previous study, we investigated the presence of B19V in archived liver tissues from patients undergoing liver transplantation for the management of ALF. B19V DNA and viral replication were detected in matched serum and explanted liver samples from 23% of the patients [14]. However, further studies are required to understand the correct meaning of the detection of B19V DNA in their serum and tissues. In this way, differentiating whether genome detection corresponds to an infectious virus could help to clarify the causal relationship between B19V infection and ALF.

These data demonstrate that a laboratory test to establish the nature of B19V DNA (infectious particle or not) can be a tool to interpret the presence of B19V DNA correctly, which would be valuable for determining a causal relationship between B19V and atypical clinical manifestations. Additionally, this simple test could be further utilized to ensure the safety of blood products for transfusion and transplantation purposes.

Thus, this study aimed to evaluate the method based on the Benzonase^®^ treatment for differentiation between the presence of the infectious virions from “naked” DNA in serum and bone marrow samples to be applied in the routine diagnosis of the B19V infection. In addition, we estimated the period of viremia and DNAemia in the sera and bone marrow of nonhuman primates experimentally infected with B19V.

## 2. Materials and Methods

### 2.1. Biological Samples and Ethical Aspects

The performance of the assay was evaluated, in duplicate, using different samples types that have been stratified into four panels, as follows:

Panel 1. Includes five archived (−20 °C) serum samples (encoded as H01 to H05) obtained from patients with exanthematic illness, sent to the LADTV/IOC-Fiocruz for B19V diagnostic evaluation. These samples were characterized as B19V DNA and anti-B19V IgM-positive and were used to establish the optimum concentration of endonuclease in serum treatments.

Panel 2. Ten follow-up serum samples were obtained from five patients with exanthematic illness (encoded as H06 to H10) who tested positive for B19V-DNA and anti-B19V IgM.

Panel 3. Ten follow-up serum samples were obtained from five patients with unspecific symptoms (encoded as H11 to H15) who tested positive for B19V-DNA and anti-B19V IgG and negative for B19V IgM.

Panels 2 and 3 included serum samples from patients monitored for up to 30 days, enrolled at the Hospital Antonio Pedro-Federal Fluminense University and Hospital Getúlio Vargas Filho (Niterói, Brazil). Blood samples from each patient were obtained at the onset of the symptoms during the medical consultation [considered as the 0-day post-admission (dpa)], and a second sample was obtained 30 days later (30 dpa) (Figure 1).

Panel 4. Since the timing of infection of the patients is unsure, we included serial serum and bone marrow samples from four cynomolgus monkeys (*Macaca fascicularis*), experimentally infected with B19V (encoded as Cy01 to Cy04) to accurately determine the time of infection and correlate it with B19V DNA levels. These animals were clinically healthy, young adults (weighing 3–5 kg), ranging in age from three to four years old from the Department of Primatology, Institute of Science and Technology in Biomodels (Fiocruz), as described by Leon et al. [18]. These samples collected at 7, 14, 19, 30, 45 and 60 days post B19V inoculation (dpi) were stored at −70 °C in the LADTV-IOC/Fiocruz biorepository until the beginning of this study [18] (Figure 1).

The study protocol was approved by the Ethics Committee of Oswaldo Cruz Institute (protocol # 1.896.353), all partnering health units, and the Animal Use Ethics Committee of Oswaldo Cruz Institute (protocol # P0064-00).

### 2.2. Diagnostic Criteria of B19V Infection

Serum samples were tested, in duplicate, for anti-B19V IgG and anti-B19V IgM by SERION ELISA classic Parvovirus B19 IgG and IgM^®^ (Virion\Serion, Wurzburg, Germany), according to the manufacturer’s instructions. These assays have sensitivity and specificity > 99% and are composed of virus-like particles (VLPs) containing recombinant VP2 protein to detect B19V IgG and IgM [19]. For anti-B19V IgM detection, a previous dilution of the samples with SERION Rheumatoid Factor-Absorbent (Virion\Serion, Wurzburg, Germany) was made to avoid non-specific ligation of IgM-antibodies (rheumatoid factors), that could lead to false-positive results.

Panels 1 and 2 include serum samples from patients with specific symptoms of B19V infection, anti-B19V IgM and IgG positive, and B19V-DNA positive, indicating a profile of acute infection. On the other hand, panel 3 includes serum samples anti-B19V IgM negative, anti-B19V IgG positive and B19V-DNA positive, indicating a profile of past and persistent infection.

### 2.3. Endonuclease Pretreatment

To investigate whether the B19V DNA detected was encapsidated in viral particles (virions) or not (“naked DNA”), serum and BM samples were pretreated with the Benzonase^®^ enzyme (25 kU; Sigma-Aldrich, San Luis, MO, USA). Benzonase^®^ is a genetically engineered endonuclease obtained from *Serratia marcescens* that degrades all forms of genetic material (DNA and RNA) that are free in the sample (i.e., not encapsidated by protein particles in the case of viruses or protected by the lipid bilayer in the case of cells) [16].

The efficiency of Benzonase^®^ to cleave viral nucleic acids present in the serum samples were evaluated as follows. Each serum sample (200 µL) was divided into two 100 µL aliquots. Benzonase^®^ was added to one aliquot, and the other aliquot was not treated with endonuclease as the control sample to compare the effect of the pretreatment.

Both aliquots (Benzonase^®^-pretreated and the control) were incubated together while shaking at 120 rpm for 1 h at 37 °C. Then, the samples were kept at room temperature to stop the reaction, as recommended by the manufacturer and FDA guidelines [20], followed by viral DNA extraction and PCR for the specific detection of B19V DNA. To determine the optimal endonuclease concentration and evaluate its effect on different B19V viral loads, a panel of five serum samples (H01 to H05) with different viral loads (10^3^ IU/mL to 10^7^ IU/mL) were analyzed, in duplicate, with the following concentrations of Benzonase^®^ (25 kU): 0.1 U/µL; 0.5 U/µL; 1.0 U/µL; 1.5 U/µL; 2.0 U/µL; 2.5 U/µL.

Positive and negative virion controls were also included in each experiment. As a positive virion control, a serum (sample H05) from a patient with acute B19V infection (anti-B19V IgM-positive and B19V-DNA of 10^7^ IU/mL, genotype 1A) was used. As a negative control of virion, a “naked” B19V-DNA (purified from a serum sample) was used. These controls were pretreated with Benzonase^®^ in the same way as the samples and tested with and without pretreatment with an endonuclease.

We used a modified MAPIA (multi-antigen print immunoassay) to evaluate if there were viral particles in serum samples in which the genome titer remained detectable after benzonase pretreatment. This assay consists of a thin layer of immobilizated antigen or antibody onto nitrocellulose membrane by micro-printing, without denaturing conditions, followed by a standard chromogenic immunoassay [21]. In this case, mouse IgG anti-B19V VP1 monoclonal (1:1.200) (Abcam Inc., Waltham, MA, USA) was micro-printed to a nitrocellulose membrane using semi-automatic printing equipment (CAMAG automatic TLC sample 4, CAMAG, Muttenz, Switzerland). The membranes were incubated (37 °C, 30 min) with 1mL of blocker buffer PBS-milk (0.5% skimmed milk (*w*/*v*) in PBS). After that, the membrane was washed (3-fold) with PBS-Tween 20 (0.05%) (PBST) and incubated (37 °C, 30 min) with serum, diluted in PBST (1:100), H09 (0 dpa; B19V-DNA negative after benzonase pretreatment) and H10 (0 dpa; B19V-DNA positive after Benzonase^®^ pretreatment). After washing (3-fold, 37 °C, 5 min) with PBST, the membranes were incubated (37 °C, 1 h) with the mouse anti-B19V VP1 monoclonal (1:1.200). After the washing step, the conjugate (Goat IgG anti-mouse with peroxidase; 1:1.200) was added and incubated (37 °C, 30 min). After an additional washing step (3-fold), the immune complex was revealed with diaminobenzidine in citrate/phosphate buffer, pH 5.0, 30% H_2_O_2_. The membranes were rinsed in purified water at room temperature to stop the reaction. The strips were dried to be analyzed.

### 2.4. B19V DNA Extraction and qPCR

B19V DNA was extracted from all samples, including the nuclease-treated and nontreated aliquots and the positive and negative controls, using a QIAamp DNA Mini Kit (Qiagen, Hilden, Germany), according to the manufacturer’s instructions. DNA was eluted with 200 µL of elution buffer and stored at −70 °C until use.

Real-time PCR (qPCR) was carried out using the TaqMan system (Applied Biosystems 7500 Real-Time PCR System, Applied Biosystems, Waltham, MA, USA) as described previously [22]. For absolute quantification, a synthetic standard curve of the B19V NS1 region [custom synthesized by IDT^®^ (CoralVille, IA)] was designed (nt 1905–1987, GenBank: NC_000883.2). Primers for the NS1 region (nt 1905F and 1987R) and a single labeled 5′ FAM probe (nt 1925–1948, GenBank: NC_000883.2) were used.

### 2.5. Statistical Analysis

Statistical analyses were performed with GraphPad Prism 8.3.1. software (GraphPad Software, San Diego, CA, USA). Continuous variables were expressed as mean and were compared with the Mann-Whitney U test. All *p*-values were two-sided, and those <0.05 were considered statistically significant.

## 3. Results

### 3.1. Determination of the Optimal Benzonase^®^ Concentration

Serum samples (H01-H05) from five patients with acute B19V infection were pretreated with Benzonase^®^, and viral DNA was then extracted from these samples and subjected to qPCR analysis.

As shown in Table 1 and Figure 2, the amount of B19V DNA in samples containing 10^5^ IU/mL (H01 and H04) was not significantly reduced after Benzonase^®^ pretreatment at concentrations varying from 0.1 to 2.0 U/µL, indicating the presence of infectious virions. However, the viral DNA load exhibited at least one log_10_ reduction in the other three samples (H02, H03 and H05) containing 10^6^ to 10^7^ IU/mL after treatment with 2.0 U/µL Benzonase^®^. The viral DNA loads remained almost the same, using a higher concentration of Benzonase (2.5 U/µL). Thus, we established a concentration of 2.0 U/µL to test the clinical samples.

The effect of Benzonase^®^ pretreatment (1.0 U/µL) on the load of the “naked” DNA controls reduced the genome copy numbers to undetectable levels, signifying that the samples comprised naked DNA.

The positive control (sample H05) was positive in all experiments, so there were virions in this sample. On the other hand, the negative control, which consists of the DNA from the serum sample treated with Benzonase^®^, was negative in all experiments.

### 3.2. Effect of Benzonase^®^ Pretreatment on the B19V DNA Levels in Serum Samples from Infected Patients

Serial serum samples were obtained from ten patients that were B19V DNA positive. Among them, five patients had a profile of B19V acute infection (H06-H10; average age of 10 years, ranging from 3 to 39 years), and five had a profile of B19V persistent infection (H11-H15; average age of 9 years, ranging from 6 to 28 years). All sera collected from anti-B19V IgM positive individuals, 30 days post-admission (30 dpa), remained positive for B19V DNA. In contrast, second samples from most of the individuals’ anti-B19V IgM negative (3/5) became negative for B19V DNA (Table 2 and Figure 3).

After pretreatment with endonuclease (2.0 U/µL), in most of the samples anti-B19V IgM positive (H06, H07, H08 and H10), the B19 DNA remained detectable, despite the DNA load being partially reduced after pretreatment, suggesting that part of the B19V DNA in these samples was encapsidated in viral particles (Table 2). On the other hand, among patients without anti-B19V IgM, three (H11, H14 and H15) became undetectable, which is compatible with a nonencapsidated state of B19V DNA. However, all the samples collected 30 days after admission were 100% sensitive to endonuclease, confirming the presence of “naked” DNA.

In both groups (anti-B19V IgM positive and negative), the differences in the viral load before and after endonuclease pretreatment were statistically significant in samples collected at 0 and 30 dpa (*p* < 0.05).

The presence of encapsidated B19V DNA in a serum sample (H10) in which the B19V DNA remained detectable after pretreatment with Benzonase^®^ was proven by MAPIA (Figure 4A). The immune complex of anti-B19V/ B19V VP1-VP2 confirmed the occurrence of B19V particles in this sample. In contrast, Figure 4B shows the absence of B19V particles in a serum sample (H09) in which B19V DNA became undetectable after pretreatment with Benzonase^®^.

### 3.3. Evaluation of B19V DNA in Serum and Bone Marrow Samples from Experimentally Infected Cynomolgus Monkeys

As shown in Figure 5, a correlation between the appearance of anti-B19V IgG in serum and the sensitivity of viral DNA in serum and bone marrow to Benzonase^®^ treatment was observed.

In the Cy01, which showed early seroconversion of IgG at the 7th dpi, B19V DNA remained detectable until the 30th dpi after benzonase^®^ treatment. While, Cy02 and Cy03, in which seroconversions were detected at the 14th dpi, B19V DNA remained detectable until the 45th dpi, indicating a longer presence of encapsidated B19V DNA in these animals with later seroconversion, than that observed in Cy01 (Figure 5).

From 30th dpi, a reduction in viral loads was seen in serum and bone marrow samples after Benzonase^®^ treatment compared with nontreated samples, indicating that “naked” and encapsidated B19V DNA coexist until the 60th dpi. At this point, the DNA from the infectious virions was no longer detectable. For Cy04, as seroconversion occurred only at the 45th dpi, B19V DNA remained detectable after endonuclease treatment, demonstrating the longest presence of infectious virions.

The difference between the treated and nontreated serum samples was not statistically significant (*p* > 0.05).

## 4. Discussion

In this study, we evaluated the performance of a method based on sample pretreatment with Benzonase^®^ followed by qPCR to detect and quantify B19V infectious viral particles (virions) in serum and in bone marrow. It has been reported that viral DNA can be diagnosed in peripheral blood several days or months after the acute infection [23,24,25]. To understand the clinical significance of long-term B19V DNA detection, reliable methods are needed to confirm the virions in tissues from these patients and their association with the B19V clinical course, as suggested by other researchers in the context of autoimmune thyroid gland diseases [26], the skin without dermatological injury [27], and myocardial inflammatory injury [28].

Serial samples from naturally B19V-infected individuals and experimentally infected cynomolgus monkeys demonstrated that the presence of B19V DNA in the blood did not necessarily correlate with the presence of infectious viral particles (virions) circulating. This finding is in accordance with a previous study that revealed the presence of virions only at the beginning (approximately 60 days) of the infection. Molenaar-de-Backer and collaborators determined that after five months, only naked-strands B19V DNA was found [16].

The ideal concentration of Benzonase^®^ for use in serum and BM samples was determined. An optimal Benzonase^®^ concentration of 2.0 U/µL was adopted since, using a higher concentration of Benzonase^®^ (2.5 U/µL), the viral loads remained the same. All serum samples from patients with and without anti-B19V IgM showed partially or completely reduced viral loads after Benzonase^®^ pretreatment, which suggested the cooccurrence of viremia and DNAemia in both groups of patients on the first collection day. Most of the patients with a profile of acute infection (H06, H07, H08, H10) remained with the viral DNA detectable, showing a partial reduction of the genome titer, which suggests that during the acute phase of infection, the DNA is predominantly encapsidated. The MAPIA showed that the detection of viral DNA in these samples, even after endonuclease treatment, could be ascribed to the presence of the encapsidated viral DNA.

The viral DNA from three patients with a profile of acute infection (H09) became undetectable after endonuclease treatment on the first collection day. This result suggests that probably this patient was in an advanced period of infection and no longer in an acute infection. This indicates that the Benzonase^®^ pretreatment assay must be used as an additional approach to attest the status of infection. Although IgM, in general, is a sensitive indicator of recent infection, it lacks clinical specificity in many contexts. Furthermore, anti-B19V IgM can remain detectable for approximately 30 days post-infection [29], or it can, in the acute phase, be of very low titer and thus remain below the threshold of detection in a low-sensitive IgM ELISA. Lastly, it can pick up non-specific IgM [30]. Approximately 30 days after the first collection, all samples were negative for B19V DNA after Benzonase^®^ pretreatment, demonstrating that there was only ‘naked’ B19V DNA. These samples collected approximately 30 days after infection showed viral loads between 10^3^–10^5^ IU/mL, which suggested that samples with a viral load ≤10^4^ IU/mL do not contain infectious virus, as suggested by other groups, European Pharmacopoeia and Food and Drug Administration [29,31,32].

The 60-day follow-up of experimentally infected cynomolgus monkeys (*Macaca fascicularis*) revealed high B19V viremia in most animals even after seroconversion. The results observed in patients and animals were similar, as the endonuclease pretreatment experiments demonstrated that during the first 30 days of B19V infection, the presence of B19V DNA was indicative of infectious virus. Subsequently, B19V DNA was degraded by the endonuclease, indicating that “naked DNA” was predominantly present. B19V “naked DNA” was generally diagnosed under the viral load of 10^5^ IU/mL, indicating that samples with this viral load are not potentially infectious.

These results are of great relevance for blood banks, considering that the detection of B19V DNA in blood donor samples with a viral load ≤10^4^ IU/mL does not pose a residual risk for blood transfusion transmission [33,34] since these blood donors with low viremia, no longer have B19V virions.

In the persistent infection, there is a lower viral genome expression, possibly contributing to the maintenance of the virus in tissues, that can be relevant to the balance and outcome of the different types of infection associated with B19V [35]. It has been proposed that the persistence of B19V DNA in blood, also known as DNAemia, can result from B19V DNA released from apoptotic or necrotic cells [36,37]. The passive release of viral DNA may likely explain the genome present in the blood in some cases, while in other cases, active viral infection is responsible. Therefore, B19V DNA present in the blood approximately 45 days after acute infection probably derives from a different origin, but it remains unknown hitherto.

B19V infects erythrocyte progenitors binding to the cellular receptor globoside (or P antigen), which induces structural changes in the capsid, leading to the accessibility of the N-terminal region of VP1 (VP1u) that is required for internalization [38]. However, this internalization in cell types that are not permissive to infection does not appear to start a productive infection cycle, leading to an accumulation of B19V DNA [38]. This was confirmed by the presence of B19V DNA in many tissues that are not permissive to the virus [13]. Naked DNA can be released into the circulation by apoptosis or necrosis or when cells are normally renewed and via exocytosis. This release by cells can occur for years or even decades after the initial B19V infection, depending on cell type. Since different cell types have different turnover rates, this can cause an aleatory release of B19V DNA, explaining the gradual decline in viremia after acute infection. [16].

Conventional and real-time molecular assays can detect low quantities of viral DNA because they are extremely sensitive techniques. However, they do not differentiate between infectious virus particles and “naked DNA.” Therefore, the treatment method with an endonuclease that cleaves any DNA or RNA present in a sample is quite useful in determining whether B19V DNA is infectious and linked to replication [16].

The availability of a simple and reliable method to distinguish viral particles from naked genetic material could be especially applicable for no cultivable viruses, such as B19V, whose productive infection is highly restricted to erythroid progenitor cells of the bone marrow [4], and for viruses whose isolation procedure requires a high biosafety level, such as SARS-CoV-2. Other future applications of this test include the correct diagnoses of several viral infections, as prolonged periods of viremia have been increasingly observed with the development of highly sensitive molecular techniques for viral genome detection and the detection of viral genomes spread in multiple organs of infected patients. Establishing the relevance of viremia in these cases and the causal relationship of this finding with infection is of paramount importance for the correct laboratory diagnoses of viral infections.

The small human (n = 10) and monkey (n = 4) sample sizes and the differential periods of observation for the collected samples (30 days for humans and 60 days for monkeys) could be considered limitations of this study.

## 5. Conclusions

The laboratory test based on Benzonase^®^ pretreatment of serum and BM samples enabled the discrimination of “naked DNA” from B19V infectious particles in these clinical specimens. Therefore, this test can be a valuable tool to be used in routine diagnosis to clarify the role of B19V as an etiological agent associated with atypical clinical manifestations and an additional approach to attest the timing of the infection.

## Figures and Tables

**Figure 1 viruses-14-00843-f001:**
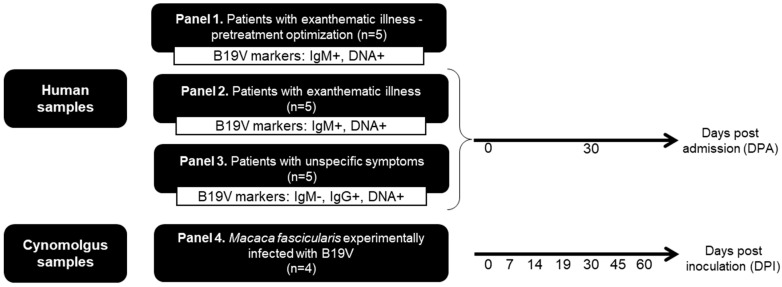
Study population. Serum samples were obtained from patients with B19V markers (IgM, IgG and DNA). Serum and bone marrow (BM) samples were obtained from cynomolgus monkeys experimentally infected with B19V.

**Figure 2 viruses-14-00843-f002:**
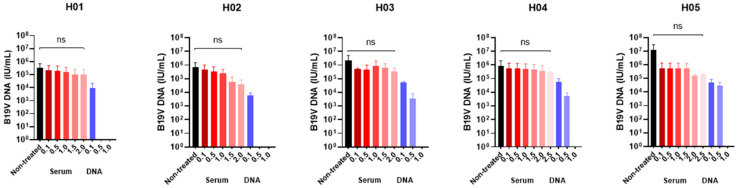
Optimization of Benzonase^®^ concentration among human serum samples. IU/mL: International Units per milliliter. Ns: non-significative.

**Figure 3 viruses-14-00843-f003:**
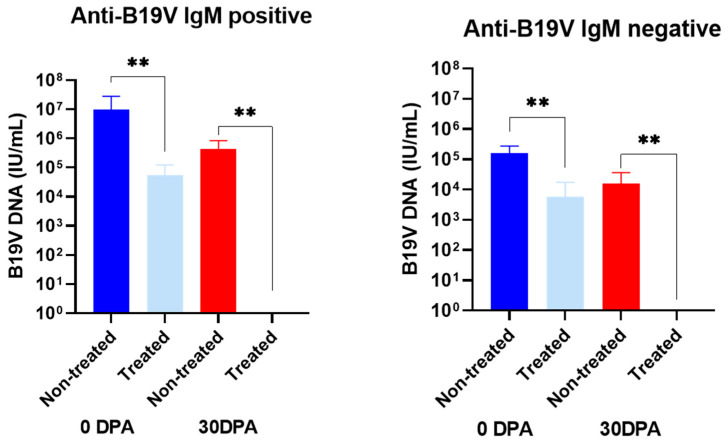
Performance of Benzonase^®^ pretreatment in serial serum samples collected from patients with anti-B19V IgM positive and negative. IU/mL: International Units per milliliter; DPA: Days post-admission; ** *p*-value < 0.01.

**Figure 4 viruses-14-00843-f004:**
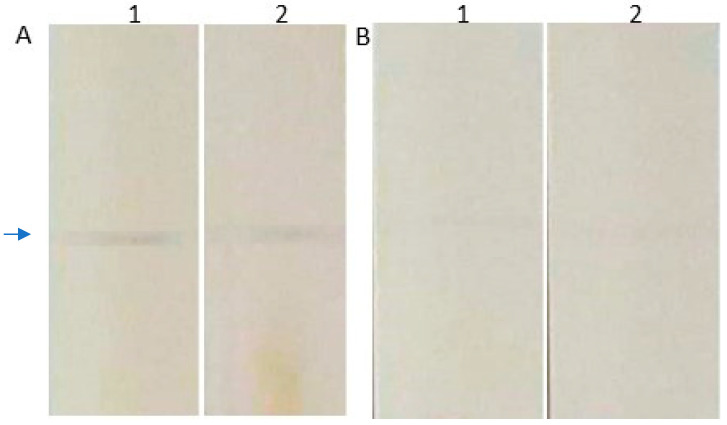
Evaluation of B19V particles’ presence in serum samples by MAPIA. In this assay, the immunocomplex of mouse anti-B19V monoclonal antibody/ B19V VP1-VP2 capsid proteins was revealed by goat IgG anti-mouse-peroxidase (arrow). (**A**) Serum sample (H10) with B19V DNA detectable before (**1**) and after benzonase pretreatment (**2**). (**B**) Serum sample (H09) with B19V DNA undetectable before (**1**) and after benzonase pretreatment (**2**).

**Figure 5 viruses-14-00843-f005:**
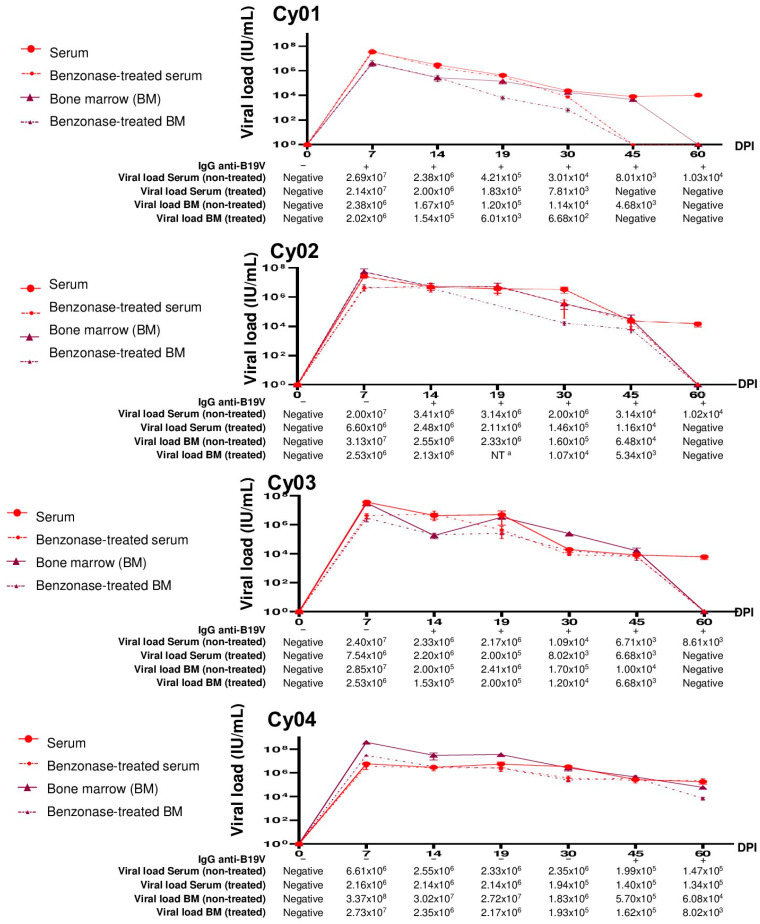
Follow-up of anti-B19V IgG and B19V DNA levels during the experimental infection in cynomolgus monkeys (n = 4). IU/mL: International Units per milliliter; BM: Bone marrow; −: Negative; +: Positive; DPI: Days-post infection; ^a^ Not tested due to insufficient bone marrow sample.

**Table 1 viruses-14-00843-t001:** Optimization of Benzonase^®^ concentration among human serum samples.

ID	Serum (IU/mL)							DNA ^a^ (IU/mL)		
	Nontreated (Mean ± SD)	0.1 (Mean ± SD)	0.5 (Mean ± SD)	1.0 (Mean ± SD)	1.5 (Mean ± SD)	2.0 (Mean ± SD)	2.5 (Mean ± SD)	0.1 (Mean ± SD)	0.5 (Mean ± SD)	1.0 (Mean ± SD)
H01	3.5 × 10^5^ ± 3.3 × 10^5^	2.1 × 10^5^ ± 2.0 × 10^5^	1.9 × 10^5^ ± 1.9 × 10^5^	1.5 × 10^5^ ± 1.5 × 10^5^	1.0 × 10^5^ ± 1.4 × 10^5^	1.1 × 10^5^ ± 1.4 × 10^5^	NT	9.4 × 10^3^ ± 8.8 × 10^3^	Negative	Negative
H02	6.6 × 10^6^ ± 5.7 × 10^6^	4.6 × 10^5^ ± 3.8 × 10^5^	3.4 × 10^5^ ± 2.7 × 10^5^	2.4 × 10^5^ ± 1.9 × 10^5^	5.6 × 10^4^ ± 6.2 × 10^4^	3.9 × 10^4^ ± 3.4 × 10^4^	NT	6.1 × 10^3^ ± 2.1 × 10^3^	Negative	Negative
H03	2.2 × 10^6^ ± 2.0 × 10^6^	4.9 × 10^5^ ± 8.9 × 10^4^	4.8 × 10^5^ ± 3.6 × 10^5^	8.9 × 10^5^ ± 7.7 × 10^5^	6.4 × 10^5^ ± 4.8 × 10^5^	3.3 × 10^5^ ± 2.1 × 10^5^	NT	5.1 × 10^4^ ± 6.4 × 10^3^	3.4 × 10^3^ ± 3.2 × 10^3^	Negative
H04	8.6 × 10^5^ ± 8.4 × 10^5^	5.8 × 10^5^ ± 6.5 × 10^5^	5.4 × 10^5^ ± 6.1 × 10^5^	4.9 × 10^5^ ± 5.7 × 10^5^	4.7 × 10^5^ ± 5.2 × 10^5^	3.8 × 10^5^ ± 4.3 × 10^5^	2.1 × 10^5^ ± 4.8 × 10^4^	5.6 × 10^4^ ± 3.4 × 10^4^	5.2 × 10^3^ ± 2.6 × 10^3^	Negative
H05	1.3 × 10^7^ ± 1.7 × 10^7^	5.4 × 10^6^ ± 6.7 × 10^6^	5.6 × 10^5^ ± 5.9 × 10^5^	5.4 × 10^5^ ± 6.0 × 10^5^	5.2 × 10^5^ ± 6.2 × 10^5^	1.5 × 10^5^ ± 3.1 × 10^4^	1.1 × 10^5^ ± 1.0 × 10^4^	4.9 × 10^4^ ± 2.9 × 10^4^	3.1 × 10^4^ ± 1.8 × 10^4^	Negative

ID: Identification; IU/mL: International Units per milliliter; SD: standard deviation; NT: Non-tested due to insufficient sample volume. ^a^ Extracted DNA was also treated with Benzonase^®^ to become a negative control and to check the standardized Benzonase^®^ concentration.

**Table 2 viruses-14-00843-t002:** Performance of Benzonase^®^ pretreatment among human sera samples from patients B19V DNA positive, according to anti-B19V IgM presence.

	ID	Sera Collected at 0 dpa			Sera Collected at 30 dpa		
		B19V Load (IU/mL) (Mean ± SD)	B19V Load Treated (IU/mL) (Mean ± SD)	*p*-Value	B19V Load (IU/mL) (Mean ± SD)	B19V Load Treated (IU/mL) (Mean ± SD)	*p*-Value
Anti-B19V IgM positive	H06	9.3 × 10^5^ ± 8.7 × 10^5^	1.6 × 10^4^ ± 1.4 × 10^4^	<0.01	1.7 × 10^5^ ± 1.4 × 10^5^	Negative	<0.01
	H07	4.3 × 10^7^ ± 3.1 × 10^7^	6.4 × 10^4^ ± 4.1 × 10^4^		2.1 × 10^5^ ± 1.9 × 10^5^	Negative	
	H08	1.9 × 10^6^ ± 2.3 × 10^6^	2.4 × 10^4^ ± 1.9 × 10^4^		9.0 × 10^5^ ± 7.8 × 10^5^	Negative	
	H09	1.7 × 10^6^ ± 1.5 × 10^6^	Negative		8.4 × 10^5^ ± 8.2 × 10^5^	Negative	
	H10	2.3 × 10^5^ ± 1.7 × 10^5^	1.7 × 10^5^ ± 1.6 × 10^5^		2.2 × 10^3^ ± 1.6 × 10^3^	Negative	
Anti-B19V IgM negative	H11	1.9 × 10^5^ ± 2.7 × 10^5^	Negative	<0.01	Negative	Negative	<0.01
	H12	1.8 × 10^5^ ± 1.3 × 10^5^	1.8 × 10^5^ ± 1.7 × 10^5^		3.7 × 10^4^ ± 2.9 × 10^4^	Negative	
	H13	3.3 × 10^5^ ± 8.4 × 10^5^	1.8 × 10^5^ ± 1.7 × 10^5^		Negative	Negative	
	H14	4.6 × 10^4^ ± 2.5 × 10^4^	Negative		Negative	Negative	
	H15	5.7 × 10^4^ ± 4.9 × 10^4^	Negative		4.0 × 10^4^ ± 3.9 × 10^4^	Negative	

D: Identification; B19V: Parvovirus B19; IU/mL: International Units per milliliter; dpa: days post-admission; SD: standard deviation.

## Data Availability

Not applicable.

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
