# Peer review of "Evaluation of Molecular Test for the Discrimination of “Naked” DNA from Infectious Parvovirus B19 Particles in Serum and Bone Marrow Samples"

_viruses, 2022, doi:10.3390/v14040843_

Round 1
Reviewer 1 Report
In this paper, use of endonuclease is described in order to differentiate between B19V-viraemia and -DNAemia. This method offers valuable evidence concerning infectivity or judgement on the role of B19V in e.g. myocarditis.
However, the method has been described before. In the paper, the own experience with this technique is depicted.
There are two reasons why I do not support pupblication of this paper:
- the findings are not new
- and most important: 4 primates have been experimentally infected with B19V, blood and bone marrow samples were taken at multiple time points in the further course for B19V DNA detection and serology. There is no additional benefit described this experiment might have. So it is ethically not justifiable.
Author Response
In this paper, use of endonuclease is described in order to differentiate between B19V-viraemia and -DNAemia. This method offers valuable evidence concerning infectivity or judgement on the role of B19V in e.g. myocarditis.
However, the method has been described before. In the paper, the own experience with this technique is depicted.
There are two reasons why I do not support publication of this paper:
- the findings are not new
AUTHOR RESPONSE: As we have mentioned in the manuscript (lines 70-77), Molenaar-de Backer and collaborators [16] recently presented this method for differentiation between B19V DNA in EDTA-plasma samples and B19V viremia. However, in the present study we evaluated the feasibility of Benzonase in different samples (serum and bone marrow) and study population. We aimed to apply this nuclease-based assay as a tool to interpret the presence of B19V DNA correctly, which would be valuable for determining a causal relationship between B19V and atypical clinical manifestations, such as ALF (which has been our major focus). In addition, we included serial serum and bone marrow samples from four cynomolgus monkeys (Macaca fascicularis), experimentally infected with B19V to accurately determine the time of infection and correlate it with B19V DNA levels.
- and most important: 4 primates have been experimentally infected with B19V, blood and bone marrow samples were taken at multiple time points in the further course for B19V DNA detection and serology. There is no additional benefit described this experiment might have. So it is ethically not justifiable.
AUTHOR RESPONSE: As we have mentioned in the manuscript (lines 133-134) the blood and bone marrow samples from animals experimentally infected were taken during a previous study, published since 2016*. Therefore, these animal samples were stored in our lab ever since, and so we updated the protocol of ethics committee of our institute to use these stored samples with this new purpose. We believe that the use of these sample could be useful to accurately determine the time of infection and correlate it with B19V DNA levels and serology, since the timing of infection of the patients included was unsure.
*Leon LA, Marchevsky RS, Gaspar AM, Garcia Rde C, Almeida AJ, Pelajo-Machado M, et al. Cynomolgus monkeys (Macaca fascicularis) experimentally infected with B19V and hepatitis A virus: no evidence of the co-infection as a cause of acute liver failure. Mem Inst Oswaldo Cruz. 2016;111(4):258-66.

Reviewer 2 Report
The manuscript by Alves et al. introduces and presents a clinical issue of parvovirus B19 genome presence for an extended period of time in clinical samples, which should be evaluated and differentiated from the presence of infectious virions, due to interference with the clinical safety of blood donation. The authors present a nuclease-based assay, which could be of use in the clinical evaluation of human blood and serum samples, when B19V transmission is of a risk.
Overall, the manuscript opens with an interesting premise and with clinical merit. Such an assay, after proper validation, might possess potential from a clinical perspective. Where the manuscript lacks, however, is experimental soundness and the interpretation of its results accordingly.
Major points:
- There are no error bars or standard deviation values presented in any of the graphs/tables, or any mention of multiplicate execution of the experiments. Each viral genome load assessment or ELISA tittering should have been performed in at least duplicates to increase the robustness of obtained results.
- There is no sound evaluation to determine that the lack of genome titer reduction following benzonase treatment could indeed be ascribed to the encapsidated nature of the viral DNA. This should have been evaluated by a capsid protein specific Western blot, ELISA or direct visualization of of viral particles by electron microscopy.
Minor points:
- taxonomy categories of both viruses and other organisms should be in Italics throughout.
- Table 1 and table 2: perhaps a histogram-based presentation of results in addition to these tables would facilitate the comprehension of these data
- Sample H07 shows a reduction of three logs in genome titer, which is similar amount of decline to what was deemed to be a sample lacking of viral particles in case of the H01 to H05 trial study. How could the authors confirm that this decline was due to the remaining fraction of DNA being packaged to particles and not due to insufficient amount of benzonase, given that this sample had the highest initial genome titer as well?
- When talking about viral load it is not clear if this means the DNA-based load or actual viral particle load
- What could explain the negative benzonase treated load on day 19 of Cy02, if it is still positive, despite of the nuclease treatment, on the following days?
- line 231: what do the authors mean by "noninfectious virions"?
- if the abbreviation B19V is introduced, it should be applied throughout the whole manuscript.
Author Response
Major points:
- There are no error bars or standard deviation values presented in any of the graphs/tables, or any mention of multiplicate execution of the experiments. Each viral genome load assessment or ELISA tittering should have been performed in at least duplicates to increase the robustness of obtained results.
AUTHOR RESPONSE: As requested for the reviewer, we repeated the experiments (Optimization of Benzonase® concentration among human serum samples) showed at tables 1 and 2, performing them in duplicates (line 103, 170). Tables 1 and 2 were changed, showing mean of viral DNA load and standard deviation values. Therefore, some parts of the topic 3.1 (lines 199-204) and of the discussion (lines 330-333) were also changed. The Elisa assays had already been performed in duplicate and this information was added in the methodology: “Serum samples were tested, in duplicate, for anti-B19V IgG and anti-B19V IgM” (line 139). We also repeated the experiments of B19V DNA quantification (pre- and post-treatment with benzonase) in samples from cynomolgus monkeys. The mean of DNA load and standard deviations values have been showed at the graphs of the figure 4, as well as the error bars.
- There is no sound evaluation to determine that the lack of genome titer reduction following benzonase treatment could indeed be ascribed to the encapsidated nature of the viral DNA. This should have been evaluated by a capsid protein specific Western blot, ELISA or direct visualization of viral particles by electron microscopy.
AUTHOR RESPONSE: As recommended, we used an assay called MAPIA (multi-antigen print immunoassay) to evaluate if there were viral particles in serum samples in which the genome titer remained detectable after benzonase pretreatment. This assay consits of a thin-layer of immobilizated antigen or antibody onto nitrocellulose membrane by micro-printing, whithout denaturing conditions, followed by standard chromogenic immunoassay. In this case, the assay was carried out to show the presence of encapsidated DNA (or viral particle) pre-and post-treatment with benzonase in a sample (H10), in which the genome titer reduced, but remained detectable, after benzonase treatment. While in a sample in which the genome became undetectable after benzonase treatment (H09), the western blot showed the absence of encapsidated virus (Figure 4A, 4B). This assay was described at methodology (lines 179-191). These results were shown at lines 299-304 and discussed at lines 369-371.
Minor points:
- taxonomy categories of both viruses and other organisms should be in Italics throughout.
AUTHOR RESPONSE: Thanks for your comment. All taxonomy categories have been rewritten in italics.
- Table 1 and table 2: perhaps a histogram-based presentation of results in addition to these tables would facilitate the comprehension of these data.
AUTHOR RESPONSE: As recommended, we included histogram-based graphs (figures 2 and 3) in addition to tables 1 and 2 to facilitate the comprehension of these data.
- Sample H07 shows a reduction of three logs in genome titer, which is similar amount of decline to what was deemed to be a sample lacking of viral particles in case of the H01 to H05 trial study. How could the authors confirm that this decline was due to the remaining fraction of DNA being packaged to particles and not due to insufficient amount of benzonase, given that this sample had the highest initial genome titer as well?
AUTHOR RESPONSE: We established 2.0 IU/mL as the ideal concentration of benzonase, since using a higher concentration (2.5 IU/mL) even in samples with high viral load, such as sample H05 (2.5x107), there was no change in viral DNA load after treatment with benzonase. In this way, we demonstrated that this concentration of benzonase was sufficient to degrade the “Naked” DNA present in sample H07, whose viral load is similar to that of sample H05. In our study, we considered a sample without viral particles, when after treatment with endonuclease there was no detection of viral DNA, and not through the reduction of genome titer after treatment. Thus, after the treatment of sample H07 with 2.0 mL of benzonase, a reduction of three logs (remaining a viral DNA load of 9.3x104) was observed due to the presence of both “naked” DNA and remaining fraction of DNA encapsidated
- When talking about viral load it is not clear if this means the DNA-based load or actual viral particle load
AUTHOR RESPONSE: We refer viral load to viral DNA-based load. Along the article, we clarified this information, changing “viral load” to “viral DNA load”
- What could explain the negative benzonase treated load on day 19 of Cy02, if it is still positive, despite of the nuclease treatment, on the following days?
AUTHOR RESPONSE: Thanks for your comment. At this day of the experimental infection (19 dpi) it was not possible to collect a sufficient volume of bone marrow from this animal to analyze the viral DNA load pre- and post-treatment with benzonase. Therefore, we changed the result to "not tested" and informed this in the figure legend (page 7). This graph has been corrected.
- line 231: what do the authors mean by "noninfectious virions"?
AUTHOR RESPONSE: Thanks for this observation. This error has been deleted.
- if the abbreviation B19V is introduced, it should be applied throughout the whole manuscript.
AUTHOR RESPONSE: As recommended by reviewer, the abbreviation B19V was applied throughout the whole manuscript.

Reviewer 3 Report
The aim of this study was to evaluate the method based on the benzonase treatment for differentiation between the infectious virions from “naked” DNA in serum and bone marrow (BM) samples to help the B19V routine diagnosis. The method may be useful; however, the manuscript needs to be revised for publication.
- Line 23, and 96, helpful?
- Line 127, -70℃
- rewrite line 140-143. It is confused.
- line 157-167, the samples were treated with Benzonase, followed by the viral DNA extraction. Will Benzonase in the samples affect the DNA extraction? How do you avoid this?
- the title of 2.3 and 2.4 are the same?
- line 213, among than?
- line 249-252, rewrite this sentence
- “naked” and encapsidated B19V DNA coexist until the 60th dpi, then, is the treatment with the Benzonase useful in clinic? Is it useful in routine diagnosis?
- the manuscript must be revised by a native English speaker.
Author Response
- Line 23, and 96, helpful?
AUTHOR RESPONSE: We changed the word “helpful” to “useful”
- Line 127, -70℃
AUTHOR RESPONSE: We corrected to -70℃
- rewrite line 140-143. It is confused.
AUTHOR RESPONSE: As suggested, this sentence was rewritten to “These assays have sensitivity and specificity >99%, and are composed by virus-like particles (VLPs) containing recombinant VP2 protein to detect B19V IgG and IgM”.
- line 157-167, the samples were treated with Benzonase, followed by the viral DNA extraction. Will Benzonase in the samples affect the DNA extraction? How do you avoid this?
AUTHOR RESPONSE: According to the manufacturer, and in agreement with the U.S. FDA guideline, the Benzonase® is only able to act as endonuclease at an optimal temperature of 37°C. So, at this optimal temperature the benzonase can cleaves the “free DNA” present in the sample.
In our study, we could control the performance of the benzonase pretreatment of serum, according to its concentration. It was showed that, in general, the pretreatment of benzonase in samples with B19-DNA load of 105-107 IU/mL led to a non-significative reduction of DNA load (Table 1 and figure 2), which is indicative that the endonuclease did not affect the DNA extraction. While, in samples with DNA load <104 IU/mL, the pretreatment with benzonase led to a partially reduction or total degradation of B19 DNA, indicating this acting at naked DNA in the sample. This result has been corroborated in previous studies that showed that the performance of the benzonase pretreatment in samples did not affect or inhibited the DNA extraction and detection (Molenaar-de Backer et al., 2016; Reber et al., 2017).
Guidance for Industry, Characterization and Qualification of Cell Substrates and Other Biological Materials Used in the Production of Viral Vaccines for Infectious Disease Indications, U.S. Department of Health and Human Services Food and Drug Administration Center for Biologics Evaluation and Research [February 2010]
Molenaar-de Backer MW, Russcher A, Kroes AC, Koppelman MH, Lanfermeijer M, Zaaijer HL. Detection of parvovirus B19 DNA in blood: Viruses or DNA remnants? J Clin Virol. 2016;84:19-23.
Reber U, Moser O, Dilloo D, Eis-Hubinger AM. On the utility of the benzonase treatment for correct laboratory diagnosis of parvovirus B19 infection. J Clin Virol. 2017;95:10-1.
- the title of 2.3 and 2.4 are the same?
AUTHOR RESPONSE: Thanks for the comment. The title 2.4 was changed to “B19V DNA extraction and qPCR”.
- line 213, among than?
AUTHOR RESPONSE: Thanks for the comment. The word has been corrected to “among them”
- line 249-252, rewrite this sentence
AUTHOR RESPONSE: As suggested, we have rewritten this sentence to clarify: “In the Cy01, which showed early seroconversion of IgG at the 7th dpi, B19V DNA remained detectable until 30th dpi, after benzonase® treatment. While, Cy02 and Cy03, which seroconversions were detected at the 14th dpi, B19V DNA remained detectable until 45th dpi, indicating a longer presence of encapsidated B19V DNA in these animals with later seroconversion than that observed in Cy01 (Figure 4).”
- “naked” and encapsidated B19V DNA coexist until the 60th dpi, then, is the treatment with the Benzonase useful in clinic? Is it useful in routine diagnosis?
AUTHOR RESPONSE: The laboratory test based on Benzonase® pretreatment of samples enabled the discrimination of “naked DNA” from B19V infectious particles. Therefore, we propose that this test can be a valuable tool to be used in routine diagnosis to clarify the role of B19V as an etiological agent associated with atypical clinical manifestations (lines 408-412). For example, the detection of B19V DNA pre and post-treatment with benzonase indicates the presence of an active infectious virus and so, can help the clinics to understand the meaning of B19V DNA detection in serum or BM in patients with undiagnosed diseases. Additionally, this approach can be a molecular method useful to differentiate between acute and persistent B19V infection, since it was shown that at acute infection virions are predominantly present, while at persistent infection the “naked” DNA is predominantly one.
- the manuscript must be revised by a native English speaker.
AUTHOR RESPONSE: As suggested, this manuscript has been revised by a native English speaker and this certificate was attached with this letter.

Round 2
Reviewer 1 Report
Sorry, I didn't realize the samples of the primates are left-over samples from a previous study. This might be more stressed, to me it sounds as if you infected new ones according to the protocol from ref. 18.
Reviewer 2 Report
The authors managed to significantly improve the quality and soundness of the manuscript.
I